# Fecal Microbiota Transplantation in Irritable Bowel Syndrome: A Systematic Review and Meta-Analysis of Randomized Controlled Trials

**DOI:** 10.3390/ijms241914562

**Published:** 2023-09-26

**Authors:** Parnian Jamshidi, Yeganeh Farsi, Zahra Nariman, Mohammad Reza Hatamnejad, Benyamin Mohammadzadeh, Hossein Akbarialiabad, Mohammad Javad Nasiri, Leonardo A. Sechi

**Affiliations:** 1Department of Microbiology, School of Medicine, Shahid Beheshti University of Medical Sciences, Tehran 1985717443, Iran or p.jamshidi@sbmu.ac.ir (P.J.);; 2School of Medicine, Shahid Beheshti University of Medical Sciences, Tehran 1985717443, Iran; 3Student Research Committee, School of Medicine, Shahid Beheshti University of Medical Sciences, Tehran 1985717443, Iran; 4Basic and Molecular Epidemiology of Gastrointestinal Disorders Research Center, Research Institute for Gastroenterology and Liver Diseases, Shahid Beheshti University of Medical Sciences, Tehran 1985717413, Iran; 5NVH Global Health Academy, Nuvance Health, Danbury, CT 06810, USA; 6St George and Sutherland Clinical School, UNSW Medicine, Sydney, NSW 2217, Australia; 7Department of Biomedical Sciences, University of Sassari, 07100 Sassari, Italy; 8SC Microbiologia e Virologia, Azienda Ospedaliera Universitaria, 07100 Sassari, Italy

**Keywords:** gastrointestinal microbiome, irritable bowel syndrome, fecal microbiota transplantation, meta-analysis, randomized controlled trial

## Abstract

Irritable bowel syndrome (IBS) poses a significant challenge due to its poorly understood pathogenesis, substantial morbidity, and often inadequate treatment outcomes. The role of fecal microbiota transplantation (FMT) in managing IBS symptoms remains inconclusive. This systematic review and meta-analysis aimed to ascertain the effectiveness of FMT in relieving symptoms in IBS patients. A thorough search was executed on PubMed/Medline and Embase databases until 14 June 2023, including all studies on FMT use in IBS patients. We examined the efficiency of FMT in reducing patients’ symptoms overall and in particular subgroups, classified by placebo preparation, FMT preparation, frequency, and route of administration. Among 1015 identified studies, seven met the inclusion criteria for the meta-analysis. The overall symptomatology of FMT-treated IBS patients did not significantly differ from the control group (Odds Ratio (OR) = 0.99, 95% Confidence Interval (CI) 0.39–2.5). Multiple doses of FMT compared with non-FMT placebo, or single-donor FMT therapy compared with autologous FMT placebo also showed no significant benefit (OR = 0.32, 95%CI (0.07–1.32), *p* = 0.11, and OR = 1.67, 95%CI (0.59–4.67), *p* = 0.32, respectively). However, a single dose of multiple-donor FMT administered via colonoscopy (lower gastrointestinal (GI) administration) significantly improved patient symptoms compared with autologous FMT placebo (OR = 2.54, 95%CI (1.20–5.37), *p* = 0.01, and OR = 2.2, 95%CI (1.20–4.03), *p* = 0.01, respectively). The studies included in the analysis showed a low risk of bias and no publication bias. In conclusion, lower GI administration of a single dose of multiple-donor FMT significantly alleviates patient complaints compared with the autologous FMT used as a placebo. The underlying mechanisms need to be better understood, and further experimental studies are desired to fill the current gaps.

## 1. Introduction

Irritable bowel syndrome (IBS) is a functional disorder of the gastrointestinal tract, marked by episodic abdominal pain and changes in bowel habits [1,2]. Despite ongoing research, the precise pathophysiological underpinnings of IBS remain elusive, with most therapeutic strategies focusing on symptomatic relief, lifestyle adjustments, and psychological interventions [3,4,5,6,7].

The gut microbiome’s role in disease prevention, early detection, diagnosis, and treatment has garnered significant attention, with potential implications for various health conditions. These conditions span infectious diseases, autoimmune disorders, cancers, chronic non-communicable diseases, and psychiatric disorders affecting virtually all human organ systems [8,9,10,11]. Additionally, the gut microbiota’s composition and balance influence gastrointestinal function [12], and studies indicate a bidirectional relationship between microbiota and gastrointestinal functional disorders, including IBS [13,14,15,16,17,18,19]. A growing body of evidence suggests that interventions like probiotics, prebiotics, synbiotics, and dietary changes can mitigate IBS symptoms [5,20,21,22,23].

Fecal microbiota transplantation (FMT) has emerged as a method for replenishing beneficial bacteria in a patient’s GI tract by introducing them from a donor’s feces, thereby restoring gut microbiota balance [24,25,26,27,28]. The efficacy of FMT has been demonstrated across various disorders, including metabolic, gastrointestinal, neurological, and neuropsychiatric conditions [29,30,31,32].

Recently, there has been a surge in randomized clinical trials (RCTs) investigating the efficacy and safety of FMT for IBS treatment, but their results are marked by inconsistencies. While some RCTs confirm the effectiveness of FMT for IBS treatment [33,34,35], others show no significant patient improvement [24,36,37,38,39]. Further compounding these challenges, there is no consensus on the optimal FMT procedure, including issues related to the use of single- or multiple-donor feces, single or repeated FMT doses, and the route of FMT administration (upper or lower GI). These disparities underscore the need for a more comprehensive systematic review and meta-analysis than those currently available [40,41,42,43,44,45,46]. Against this backdrop, the authors aim to evaluate the existing RCT studies concerning the efficacy, safety, and various aspects of FMT as a treatment option for IBS.

## 2. Methods

This review conforms to the “Preferred Reporting Items for Systematic Reviews and Meta-Analyses” (PRISMA) statement [47]. PROSPERO registration ID: CRD42023452977.

### 2.1. Search Strategy and Selection Criteria

A comprehensive search was performed in PubMed/Medline and Embase databases, including all publications up to 14 June 2023. The authors did not consider any specific limitation for the starting date of the search. The search terms “Fecal Microbiota Transplantation” OR “Fecal Microbiota Transplant” OR “Fecal Microbiome Transplantation” OR “Fecal Transplant” OR “Donor Feces Infusion” OR “FMT” OR “Intestinal Microbiome Transplant” OR “Intestinal Microbiota Transfer” OR “Intestinal Microbiota Transplantation” OR “Intestinal Microbiome Transplantation” OR “Intestinal Microbiota Transplant” OR “Intestinal Microbiome Transfer” OR “Fecal Microbiota Transfer” OR “Fecal Transplantation” were combined with “Irritable Bowel Syndrome” OR “Irritable Colon” OR “IBS” OR “Mucous Colitides” OR “Mucous Colitis” to obtain the final search strategy. A detailed search strategy in each database is available in the Appendix A.

### 2.2. Study Selection

All search results were organized in EndNote X9 (Thomson Reuters, New York, NY, USA). After removing duplications, two independent reviewers (ZN and BM) assessed the title and abstract of the articles to evaluate their eligibility for inclusion in the final analysis. The full text of potentially eligible records was retrieved and separately evaluated by two other reviewers (YF and PJ). In each step, controversies were discussed with a third reviewer (MJN). The inclusion criteria were: 1. Studies were designed as prospective, randomized, placebo-controlled trials. 2. Studies included adult (aged ≥18 years) IBS patients according to the case definition of the Rome III criteria. 3. Studies compared the outcomes of FMT administration with placebo or autologous FMT administration. 4. Consideration of IBS symptom relief as the direct result.

Review articles, duplicate publications, letters, observational studies, animal studies, and studies with insufficient information about patients’ characteristics and outcomes were excluded. There were no language restrictions. Non-English articles were translated by using the Google Translate tool.

### 2.3. Data Extraction

PJ designed a data extraction form, and two independent reviewers (ZN and BM) extracted the following items from the full text of the eligible publications: first author’s name, year of publication, the country where the study was executed, type of study, study population, gender, mean age, comorbidity(ies), co-medication(s), diet modification, IBS criteria, subtype, and severity, donors’ selection criteria, FMT and placebo preparation, FMT route of administration, dosing frequency and duration, follow-up duration and technique, primary and secondary outcomes of the study, adverse events, intestinal microbiota modifications, and the number of case and control patients with and without outcome. The data were jointly reconciled, and any disagreement between the two reviewers was resolved by obtaining a third opinion (PJ).

### 2.4. Quality Assessment

The Cochrane risk of bias tool was used to assess the risk of bias in included articles based on random sequence generation, allocation concealment, blinding of participants and personnel, blinding of outcome assessment, incomplete outcome data, selective reporting, and other aspects of bias evaluation [48].

### 2.5. Meta-Analysis

Statistical analyses were performed with Comprehensive Meta-Analysis software, version v3.7z (Biostat Inc., Englewood, NJ, USA). Meta-analysis was used to compare the outcomes of FMT with the control group. The pooled efficacy rates with 95% confidence intervals (CI) were assessed using the fixed or random effects model due to the estimated heterogeneity of the actual effect sizes. The between-study heterogeneity was assessed by Cochran’s Q and the I^2^ statistic. Subgroup analysis stratified by the FMT preparation, frequency, and route of administration of the studied population was performed to minimize heterogeneity. Publication bias was evaluated statistically using Begg’s tests (*p* < 0.05 was considered indicative of statistically significant publication bias) and funnel plots [49].

## 3. Results

In the primary search in the mentioned databases, a total number of 1015 articles were found. After removing duplicated results and screening the titles and abstracts, twenty-five papers were considered for full-text data screening. Finally, seven articles were eligible for inclusion in the meta-analysis. The detailed process and exclusion reasons are noted in the PRISMA diagram in Figure 1.

### 3.1. Quality Assessment of Included Studies

The Cochrane tool for experimental studies showed that most included studies had a low risk of bias in all evaluated items. Only one study [24] needed clarification regarding allocation concealment, blinding of participants and personnel, and blinding of outcome assessment. The results of the risk of bias assessments are summarized in Table 1.

### 3.2. Characteristics of the Included Studies and Populations

The studies included in our analysis were all randomized controlled trials originating from a diverse range of countries: one each from Norway, Belgium, Denmark, Finland, and Sweden, and two from the United States. These studies collectively assessed a total of three hundred and twenty-seven patients across both case and control groups, with the mean age being 39.6 years. Most participants were women, accounting for 57.79% of the total. All patients under consideration had moderate to severe IBS, as per the IBS-SSS scoring, and the IBS-Diarrhea subtype was the most common. While data on patients’ co-existing conditions and concurrent medications were limited, the available reports suggested that most patients had no severe comorbidities and only took medicines for symptom relief. Table 2 summarizes the essential characteristics of the included studies.

The patients were followed for a mean duration of eight months, and the safety and efficacy of the FMT were assessed utilizing IBS-QoL and IBS-SSS score questionnaires.

There were different protocols for FMT preparation used in the included studies. However, the main steps were collection of the fresh stool samples from donors, homogenization with sterile 0.9% saline, filtration to remove larger particles, mixing with glycerol as a cryoprotectant, and freezing the suspension until use. In most of the studies, multiple-donor FMT was applied for the case group of patients and was prepared by mixing the processed fecal samples from different donors into one batch before being encapsulated [34,35,37]. However, in one study, fecal sample filtrates from six different donors were encapsulated separately and delivered to the patients simultaneously [24].

FMT was delivered as fresh feces in three studies [35,36,38]; meanwhile, other studies focused on frozen feces as an administrative route [24,37,39]. In one other study [34], the case group received both fresh and frozen FMT. Regarding the control groups, four of the seven included studies administered autologous FMT as a placebo [34,35,38,39], and non-FMT placebo agents were applied in the rest [24,36,37]. Most of the included studies administered the FMT medication via oral capsules or nasojejunal probe [24,35,36,37], while others (three out of seven studies) administered the FMT via colonoscopy [34,38,39].

Single-dose rather than repeated dose administration was more employed as the delivery route [24,35,37,38,39]; additionally, 30 mg or lower transmission was the most tested quantity among the different amounts of FMT delivered. The most common adverse events during FMT therapy were abdominal pain, nausea, diarrhea, and bloating. There have been no reports of severe or critical adverse events to date. Table 3 compares the interventions and control design among the included studies.

The intestinal microbiota modifications at the genus level are summarized in Table 4. Some studies reported alterations in several bacterial groups, and the overall biodiversity was increased after FMT. The most common alteration in bacterial composition in IBS patients who received FMT was the enrichment of the phyla Bacteroidetes and Firmicutes. Additionally, one study [36] found an increase in the Bacteroidetes to Firmicutes ratio.

### 3.3. Meta-Analysis

The meta-analysis demonstrated an overall pooled odds ratio of 0.99 (95%CI 0.39–2.5, I^2^: 69.9%) among FMT-treated IBS patients compared with the control group, indicating no efficacy of FMT in the IBS treatment (Figure 2).

According to the high heterogeneity of the included studies regarding placebo preparation (non-FMT placebo or autologous FMT), FMT preparation (single donor or multiple donors, frozen or fresh FMT), frequency (single or multiple doses), and the route of administration (upper or lower GI), a subgroup analysis was conducted (Table 5 and Appendix A)).

The main difference between upper and lower administration is based on the effect of the pre-colon delivery process. An oral FMT capsule administered through the upper GI route is usually subjected to some modulations due to passing through the digestive tract before delivery to the colon; on the other hand, delivery via colonoscopy, a lower GI route, bypasses these pre-delivery changes and has direct effects on colonic sites. Delivery via nasojejunal probe is carried out through the upper GI; however, it avoids the pre-processing effects; therefore, the study of Holvoet et al. [35], in which the effect of nasojejunal-delivered FMT was evaluated, was considered as a lower GI FMT delivery in our subgroup analysis.

The analysis showed that the lower GI (via colonoscopy) use of a single dose of FMT significantly improved patient complaints (odds ratio (OR) = 2.2, 95%CI (1.20–4.03), *p*-value = 0.01, I^2^ = 0.00%) compared with the autologous FMT used as the placebo. Also, the upper GI use of frozen FMT as an oral capsule product worsened patients reported symptoms significantly compared with the non-FMT placebo (OR = 0.30, 95%CI (0.13–0.68), *p*-value = 0.04, I^2^ = 29.78%).

The administration of multiple doses of FMT, compared with non-FMT placebos, had no significant efficacy (OR = 0.32, 95%CI (0.07–1.32), *p*-value = 0.11, I^2^ = 61.46%). Single-donor FMT therapy had a positive but insignificant effect on patients’ symptoms compared with those who received autologous FMT as the placebo (OR = 1.67, 95%CI (0.59–4.67), *p*-value = 0.32, I^2^ = 33.46%). Multiple-donor FMT patients significantly improved compared with the autologous FMT (OR = 2.54, 95%CI (1.20–5.37), *p*-value = 0.01, I^2^ = 0.00%). In contrast, the administration of multiple-donor FMT, in comparison to non-FMT placebo, had a negative effect in alleviating the IBS symptoms (OR = 0.16, 95%CI (0.05–0.48), *p*-value = 0.00, I^2^ = 0.00%), indicating the potential role of the type of placebo preparation in the outcome of the trials. There was no evidence of publication bias (Begg’s test *p* > 0.05) (Figure 3).

## 4. Discussion

Our results indicate that, overall, FMT treatment did not significantly improve the symptoms of IBS patients compared with the control group. However, a single dose of multiple-donor FMT administered via colonoscopy significantly improved patient complaints compared with the autologous FMT used as the placebo. Single-donor FMT therapy also had an insignificant positive effect on patients’ symptoms compared with those who received autologous FMT as the placebo.

To date, several meta-analysis studies have been published regarding the efficacy and safety of FMT on IBS patients [40,41,42,43,44,45,46]; however, the present study has several unique aspects compared with previous studies. First, our study included the most recent RCT [24], which was not included in the last similar meta-analysis [42]. Second, we did not limit our study to invasive or non-invasive FMT administration routes and considered both, contrary to the most recent systematic review study [42]. Third, in contrast to some meta-analysis studies [42,44,45], we considered the type of IBS diagnosis criteria as part of our inclusion criteria. This was due to the discrepancies between the Rome III and Rome IV criteria regarding the severity and stability of IBS symptoms; the natural history of Rome IV-defined IBS criteria is more severe than that of Rome III, and this may act as a fundamental confounding factor in treatment trials of IBS [50,51]. Our primary search found few trials with Rome IV-defined patients. Thus, we limited our inclusion criteria to Rome III.

Fourth, we conducted the subgroup meta-analysis based on different aspects, including the FMT preparation (single-donor FMT vs. multiple-donor FMT), and found a significant superiority of multiple-donor FMT over single-donor FMT compared with the autologous FMT used as a placebo, a finding which has not been demonstrated in previous meta-analysis studies. Fifth, we also evaluated the intestinal microbiota modifications reported in our included RCTs, which has not been presented in any of the previous meta-analysis studies [40,41,42,43,44,45,46].

Furthermore, contrary to previous studies, we delved deeper into the controversy between single- and multiple-FMT dose administration, which is very helpful in choosing the best approach to FMT therapy. Another notable aspect of our study is revealing the potential role of the type of placebo preparation in the outcome of the trials. According to our analysis, while multiple-donor FMT had a significant positive effect compared with the autologous FMT as the control group, a significant negative impact in alleviating the IBS symptoms was found when a non-FMT placebo was considered as the control group [34,35,37,39]. Thus, we considered unifying the control groups among the included studies to decrease the heterogenicity and improve the reliability and validity of the findings.

IBS patients have lower diversity and richness in their intestinal bacterial profile (dysbiosis) than the healthy population [52]. Evidence shows that a decrease in abundance of *Actinobacteria* spp., *Bifidobacterium* spp., and *Lactobacillus* spp. in the intestinal bacterial profile is related to IBS symptoms, and increases in *Proteobacteria* spp. and *Escherichia coli* have the same effects [53]. Dysbiosis is a crucial element for investigation in further studies. The role of *Archaebacteria* spp. and viruses should be evaluated in IBS patients who do not respond to common IBS treatments [36,54].

The effectiveness of microbiota-targeting interventions in IBS patients has been investigated in recent years [55]. Two main mechanisms justify the use of microbiome-derived interventions in IBS; the first is attributable to the local effects of microbiota in the gut mucosal environment, and the second is related to the modulatory effects of microbiota in the gut–brain axis [55]. Restoring and maintaining the gut mucosal permeability, changing the microbial composition, and soothing inflammatory cytokine release in gut mucosa are the main mechanisms of favorable local effects of gut microbiota in IBS patients [23,56,57].

Although the efficacy of probiotics in relieving IBS symptoms has been widely accepted [58,59], the efficacy of FMT-delivered microbiota is controversial [60]. While several studies support its safety and effectiveness, others do not [61,62,63,64,65,66]. Thus, we conducted a subgroup meta-analysis study to elucidate some of the current controversies regarding the different features of FMT therapy in IBS patients.

Two ways are suggested to deliver microbiota via FMT: upper GI (oral), and via lower GI (colonoscopy and nasojejunal probe). Selection of the most suitable route is controversial. In two studies by El-Salhy et al., performing FMT via the upper GI was preferred because of the reduced need for preparations for the procedure and the possibility of fresh feces administration [33,67]. Some disadvantages of upper delivery of transplant are bacterial overload in non-favorable parts of the digestive tract and procedure-related complications such as abdominal cramps, belching, and nausea, although these are transient [36,54].

On the other hand, some studies preferred delivering transplants via the lower GI, for example, by colonoscopy. The benefits of performing FMT via colonoscopy is additional bowel cleansing, which is itself helpful as a preparation of the colon environment for transplant engraftment. Hence, control groups may also benefit from bowel cleansing with symptom relief [39,53]. Our analysis showed that receiving grafts in a single dose and via colonoscopy is more effective in improving the symptoms caused by IBS.

Determination of FMT dosage is another area of great controversy; FMT with high doses or repeated transplants seems more effective for improving patients’ social functioning and providing symptom relief [52]. However, our analysis indicated that a single dose of FMT showed a significant superiority over multiple-dose FMT therapy. This outcome could be attributed to various factors. Firstly, the oral capsule is often the most accepted route for multiple FMT administrations over consecutive days due to its high tolerance [68,69], while multiple administrations via invasive delivery (e.g., colonoscopy) may lead to patient discomfort and adverse events [70,71]. Thus, in the studies of Halkjær et al. [37] and Aroniadis et al. [36] included in our systematic review, FMT was orally prescribed repeatedly for periods of twelve and three days, respectively.

Conversely, among the single-delivery studies, colonoscopy was the most popular FMT administration route [34,38,39]. Bowel cleansing before colonoscopy has been shown to enhance outcomes, while in the oral route, capsule delivery to the colon may be impeded by various secretions and materials [42]. Therefore, the administration route may influence the advantage of single-dose over multiple-dose FMT.

Secondly, repeated doses are typically prescribed for IBS patients with severe clinical presentations [52]. El-Salhy et al. utilized repeated/high-dose transplants for patients with higher Birmingham IBS symptom scores [52]. Although all studies in this systematic review targeted patients with severe IBS (IBS-SSS score ≥ 175), confounding factors like dietary adherence, genetic predisposition, and physician follow-up were not fully elucidated. They may have impacted patients’ clinical conditions. Hence, the lower rates of symptom relief observed in multi-FMT prescription studies may be attributed to the participants’ deteriorated conditions.

Various hypotheses may justify the superiority of a single transplant over multiple transplants. FMT composition is not constant and varies from donor to donor; therefore, transferring a microbiota containing bacteria with a short doubling time may be more effective than repeatedly transferring a microbiota with a long generation time [72]. Further, the compatibility of transferred FMT with the host microbiome may influence the success of the outcome, independent of the frequency of microbiome delivery. Non-responder patients may remain unresponsive even with repeated FMT, but they may benefit from receiving a transplant from another donor [34,35]. The literature has recommended FMT transmission from a close relative [73] as the recipient’s mucosal immune system might exhibit greater tolerance towards the donor’s microbiota [74].

Consequently, a single transfer of a more compatible FMT might be more beneficial than multiple transfers of a mismatched microbiome. Hence, the trend toward personalized FMT therapy based on stool culture analyses is increasingly emphasized [74]. However, IBS is a chronic disease with fluctuating long-term manifestations [1], and FMT prescription per flare episode seems to be necessary for achieving long-lasting therapy [35]. These considerations highlight the dilemma surrounding the frequency of FMT transmission, and more clinical trials are needed to assist in choosing between single or multiple administrations as the most suitable therapeutic approach.

FMT is prepared as either fresh or frozen products. Frozen transplants are easily transportable, but the diversity of their contents may be altered. Factors such as variety in composition, storage, diversity, and the stable combination of stool microbes can affect the efficacy of FMT [35,36]. The more similar the mixture of the donor’s sample is to the recipient’s, the higher the likelihood of a successful transplant [35]. Interestingly, our analysis showed that patients receiving frozen microbiota capsules experienced more severe symptoms than those receiving non-FMT placebo capsules. Due to insufficient data, we could not compare the effects of fresh versus frozen FMT. Further trials are needed to investigate this difference.

## 5. Limitations

Our study is subject to several limitations. Firstly, the reviewed articles exhibited limitations, including concurrent psychological disorders, variations in diet, unspecified co-medications, and other factors that could affect the patients’ microbiota composition. These factors pose significant challenges in generalizing our study findings. For instance, patients who have previously undergone medical therapy for IBS might have a higher likelihood of FMT failure [33,67]. Also, diet modifications, which play a crucial role in IBS treatment by affecting the balance of short-chain fatty acids (SCFAs) in the digestive tract, can potentially influence the success of FMT [33,67].

Secondly, the lack of detailed information regarding the dosing of FMT, the state of FMT preparation (fresh or frozen), and the duration of follow-up in the studies limited our ability to perform a subgroup analysis and establish the optimal conditions for each variable. Thirdly, the studies differed in their placebo choice; some used autologous FMT, while others used a non-FMT placebo for the control group. The effect of these variations on the outcomes could not be comprehensively analyzed due to limited data, underscoring the need for further trials to clarify this issue. Fourthly, there were some missing data regarding the inclusion and exclusion criteria for selection of donors, which may have influenced the generalizability of our outcome. We highlight the need for a comprehensive standard protocol for donor selection to be considered in future trials.

Lastly, our analysis predominantly included moderate to severe IBS patients, as per the IBS-SSS scoring system, thereby limiting the generalizability of our conclusions. Future trials should also consider including mild IBS cases to assess FMT’s efficacy within this patient subset.

## 6. Conclusions

Administering a single dose of multiple-donor FMT via the lower GI route significantly improves patient symptoms compared with using autologous FMT as a placebo. The results of the current study require further elucidation. Future studies involving larger patient populations, consideration of potential confounding factors, and extended follow-up periods are recommended to bridge the existing knowledge gaps.

## Figures and Tables

**Figure 1 ijms-24-14562-f001:**
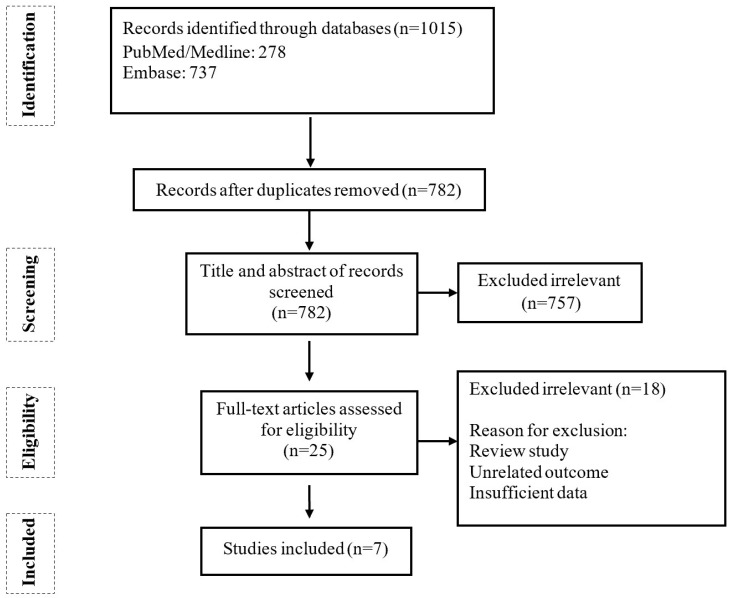
PRISMA flow chart of study selection for inclusion in the systematic review.

**Figure 2 ijms-24-14562-f002:**
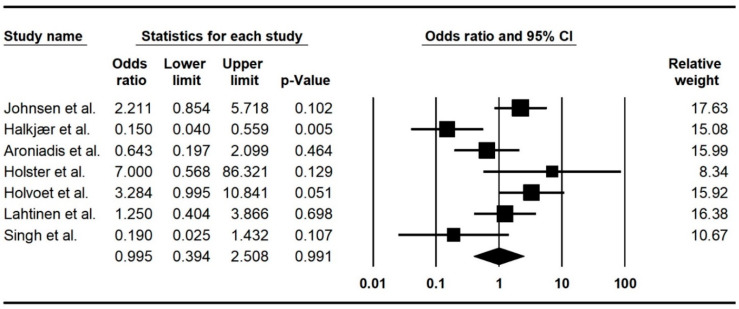
Forest plot of the included studies indicating the efficacy of FMT in IBS patients. Johnsen et al. [34]; Halkjær et al. [37]; Aroniadis et al. [36]; Holster et al. [39]; Holvoet et al. [35]; Lahtinen et al. [38]; Singh et al. [24].

**Figure 3 ijms-24-14562-f003:**
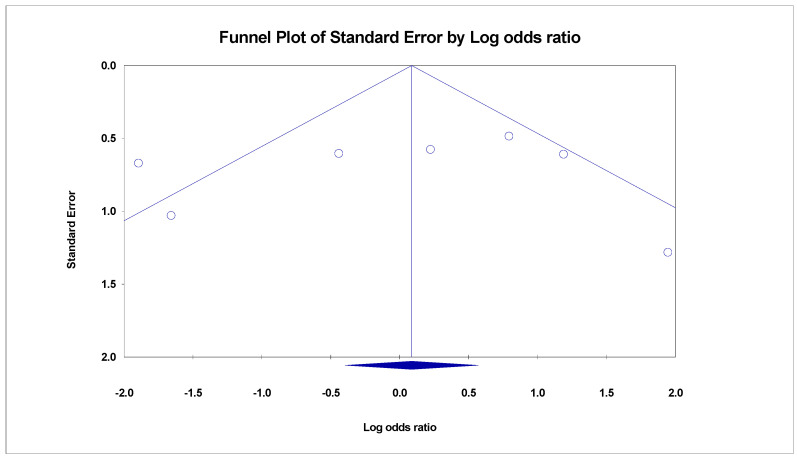
Funnel plot indicating publication bias of the selected studies.

**Table 1 ijms-24-14562-t001:** Risk of bias assessment of the experimental studies included in the meta-analysis via the Cochrane risk of the bias assessment tool.

Author	Random SequenceGeneration	Allocation Concealment	Blinding of Participants and Personnel	Blinding of Outcome Assessment	Incomplete OutcomeData	Selective Reporting	Other Bias
Johnsen et al. [34]	Low risk	Low risk	Low risk	Low risk	Low risk	Low risk	Low risk
Halkjær et al. [37]	Low risk	Low risk	Low risk	Low risk	Low risk	Low risk	Low risk
Aroniadis et al. [36]	Low risk	Low risk	Low risk	Low risk	Low risk	Low risk	Low risk
Holster et al. [39]	Low risk	Low risk	Low risk	Low risk	Low risk	Low risk	Low risk
Holvoet et al. [35]	Low risk	Low risk	Low risk	Low risk	Low risk	Low risk	Low risk
Lahtinen et al. [38]	Low risk	Low risk	Low risk	Low risk	Low risk	Low risk	Low risk
Singh et al. [24]	Low risk	Unclear	Unclear	Unclear	Low risk	Low risk	Low risk

**Table 2 ijms-24-14562-t002:** Characteristics of the included studies and patients.

First Author	Year of Publication	Type of Study	Country	Study Population	Gender Male/Female	Mean Age (Years)	Comorbidities	Co-Medications	Diet Modification	IBS Criteria	IBS Subtype	IBS Severity *	Donor Selection Criteria
Johnsen et al. [34]	2017	Randomized, double-blinded, controlled clinical study	Norway	83	28/55	44.33	Fibromyalgia, chronic fatigue syndrome, jaw pain, pelvic pain, anxiety, and depression	NA	Low FODMAPs	Rome III	IBS-D, IBS-M	≥175	Exclusion criteria: Use of antibiotics in the past 3 months; new tattoos or piercings in the past 3 months; high-risk sexual behaviors; former imprisonment; or history of any of the following conditions: chronic diarrhea, constipation, inflammatory bowel disease, IBS, colorectal polyps or cancer, immunosuppression, obesity, metabolic syndrome, atopic skin disease, or chronic fatigue. Tests for parasites, ova, and cysts; *Salmonella* spp., *Shigella* spp., *Campylobacter* spp., *Yersinia* spp., and toxin-producing *C-difficile*; fecal tests for *Helicobacter pylori* antigen, viruses (*Norovirus, Rotavirus, Sapovirus, Adenovirus*), calprotectin, and occult blood; blood samples for glycated hemoglobin; and serology for HIV, *Treponema pallidum*, and hepatitis A, B, and C.
Halkjær et al. [37]	2018	Randomized, double-blinded, controlled clinical study	Denmark	46	16/30	36.39	Asthma, allergies	PPI	NA	Rome III	IBS-D, IBS-M, IBS-C	≥175	Inclusion criteria: Aged between 18 and 45 years; previously and currently healthy; normal weight (body mass index (BMI) between 18.5 and 24.9 kg/m^2^); normal bowel movements (defined as 1–2 per day and type 3–4 at Bristol Stool Form Scale); no medication consumption. Exclusion criteria: Known or high risk of infectious diseases such as HIV, HAV, HBV or HCV; positive stool sample for *C. difficile* toxin, parasites or other enteropathogens; antibiotic treatment in the past 6 months; abuse of alcohol or drugs; smoking; tattoo or body piercing within the last 6 months; allergy, asthma or eczema; family history of GI diseases, cancer, diabetes, obesity, autoimmune diseases, allergy, asthma, eczema, cardiovascular diseases, neurologic or mental illnesses; participation in high-risk sexual behaviors; born by caesarean section.
Aroniadis et al. [36]	2019	Randomized, double-blinded, controlled clinical study	USA	48	30/18	37.3	NA	NA	NA	Rome III	IBS-D	≥175	NA (A non-profit stool bank (OpenBiome, Somerville, MA, USA))
Holster et al. [39]	2019	Randomized, double-blinded, controlled clinical study	Sweden	16	8/8	36.5	NA	Participants were asked to keep their medication stable	Participants were asked to keep their diet stable	Rome III	IBS-D, IBS-M, IBS-C	≥175	Exclusion criteria: Current communicable diseases; known organic gastrointestinal disease; gastrointestinal malignancy or polyposis, history of major gastrointestinal surgery; eosinophilic disorders of the gastrointestinal tract; known or high risk of infectious diseases such as HIV or hepatitis; non-gastrointestinal malignancy; dementia, severe depression or major psychiatric disorder; metabolic syndrome; autoimmune diseases; allergies; chronic pain syndromes; severe or morbid obesity; pregnancy or breast-feeding; use of immunosuppressive or chemotherapy agents; antimicrobial treatment within last 6 months; abuse of alcohol or drugs; tattoo or body piercing obtained within the 6 months before screening; high-risk sexual behaviors; travelling to areas with endemic diarrhea during 3 months before screening; positive stool tests for *Clostridium difficile* toxin, enteral pathogens (*Salmonella*, *Shigella*, enteroinvasive *E. coli*, *Campylobacter*, enterohaemorrhagic *E. coli*, enterotoxigenic *E. coli*, *Yersinia enterocolitica*, *Vibrio*, and *Plesiomonas shigelloides*), ova, parasites, *Giardia* antigen, *Cryptosporidium* antigen; positive blood tests for HIV, Hepatitis A, B, or C.
Holvoet et al. [35]	2020	Randomized, double-blinded, controlled clinical study	Belgium	62	24/38	38.7	NA	NA	NA	Rome III	IBS-D, IBS-M	≥175	Inclusion criteria: Being in good overall condition, between 18 and 65 years of age, with normal, regular bowel movements and no gastrointestinal symptoms. Exclusion criteria: Body mass index (BMI) > 30, antibiotic use in the past 6 months, chronic disease or abnormal screening results. Donors were subjected to a clinical examination at the start of the trial and were screened for various transmittable diseases at six-month intervals. Screening tests included testing for hepatitis A, B, C and E, HIV-1 and 2 and *Treponema pallidum*; stool culture for the presence of *Salmonella spp.*, *Shigella spp.*, *Yersinia enterocolitica*, *Yersinia pseudotuberculosis*, *Campylobacter spp.*, *Clostridioides difficile* and *Aeromonas spp.* Additionally, specific screening for antibiotic-resistant strains was performed using the active detection of carbapenemase-producing Enterobacterales (CPE) and extended spectrum beta-lactamase (ESBL) producing organisms. Microscopic examination was performed to confirm the absence of eggs, cysts and/or larvae of parasites, and the presence of *Clostridioides difficile* toxins A and B was screened using an enzyme immuno-assay
Lahtinen et al. [38]	2020	Randomized, double-blinded, controlled clinical study	Finland	49	20/29	46.76	NA	NA	NA	Rome III	IBS-D, IBS-M	≥175	Inclusion criteria: Being in good general health and normal weight; delivered through vaginal childbirth; not having antibiotics during the previous year; not being a health care worker. Exclusion criteria: Having any long-term diagnoses; using any permanent medications; a history of high-risk sexual behavior, use of illicit drugs or recently travelled to areas with a high incidence of infectious diarrhea. The donors were screened with the following diagnostic tests: HIV; hepatitis A, B and C; culture of fecal bacterial pathogens (*Salmonella*, *Yersinia* and *Campylobacter*) and antibiotic-resistant bacteria (MRSA, ESBL); detection of *Clostridioides difficile* toxin; *Helicobacter pylori* and fecal parasites (ova and protozoa)
Singh et al. [24]	2021	Randomized, placebo-controlled, single-center study	USA	23	12/11	37.4	NA	IBS medications	NA	Rome III	IBS-D	≥150	NA (stool bank (OpenBiome, Somerville, MA, USA))

* According to the IBS-SSS score system; RCT: Randomized Controlled Trial; IBS-D: IBS-diarrhea dominant, IBS-M: IBS-mixed type, IBS-C: IBS-constipation dominant; FODMAPs: Fermentable oligosaccharides, disaccharides, monosaccharides, and polyols; NA: not available.

**Table 3 ijms-24-14562-t003:** Characteristics of the RCTs and outcomes.

First Author	FMT Preparation	Placebo Preparation	FMT Route of Administration	FMT Frequency & Dosing	FMT Duration	Follow-Up Duration	Follow-Up Technique	Primary Outcome	Secondary Outcome	Adverse Effects	Efficacy of FMT (Yes/No)
Johnsen et al. [34]	Multiple donors’ feces (frozen and fresh)	Autologous	Colonoscope	50–80 g of feces, single dose	One day	12 months	Self-assessment questionnaires (IBS SSS, fatigue, and quality of life)	Symptom relief of more than 75 points assessed by IBS-SSS	The primary outcome was reassessed at 12 months after FMT for the secondary endpoint.	Abdominal pain, nausea	Yes
Halkjær et al. [37]	Multiple donors’ frozen feces	Saline, glycerol, and food coloring E150, 30% glycerol	Oral capsules	300 g of feces daily (25 capsules daily)	12 days	Six months	Change in IBS-QoL	Reduction in IBS-SSS in the treatment group compared with the placebo group	Change in IBS-QoL scores at three months and changes in microbiota diversity before and after FMT.	Abdominal pain, nausea, diarrhea, constipation, bloating, vomiting, fatigue, fever	No
Aroniadis et al. [36]	Donor whole fresh stool	Non-toxic brown pigment	Oral capsules	28 g, 75 FMT capsules (25 capsules everyday)	Three days	Six months	Questionnaire, sequencing 16S rRNA gene with Illumina Miseq technology	Differences in IBS-SSS between the groups	The IBS-QOL questionnaire, Hospital Anxiety and Depression Scale (HADS), Bristol Stool Form Scale (BSFS), microbiome profiles as assessed by 16S rRNA sequencing, and adverse events.	Abdominal pain, nausea, diarrhea, constipation, bloating, vomiting, fatigue, belching, loss of appetite	No
Holster et al. [39]	Single-donor feces (frozen)	Autologous	Colonoscope	30 g, single dose	One day	Six months	GSRS-Ib, IBS-SSS questionnaire	Effect of FMT on IBS symptoms	IBS symptoms using the IBS-SSS, their general health and quality of life, and IBS-QoL, anxiety, and depression status	Abdominal pain, nausea, diarrhea, constipation, bloating	No
Holvoet et al. [35]	Two donors’ fresh feces	Autologous	Nasojejunal probe	300 mL of the donor solution, single dose	One day	One year	IBS-related symptoms were assessed using a daily symptom diary, IBS-QoL questionnaire, sequencing 16S rRNA gene with Illumina Miseq technology	Relief of general IBS symptoms and abdominal bloating	Changes in daily assessed IBS symptoms, quality of life, changes of fecal microbiota composition, 4 IBS-related symptoms one year following FMT.	NA	Yes
Lahtinen et al. [38]	One donor’s fresh feces	Autologous	Colonoscope	30 g, single dose	One day	One year	IBS-related symptoms were assessed using a daily symptom diary, IBS-QoL questionnaire, sequencing 16S rRNA gene with Illumina Miseq technology	Decline in the IBS-SSS score of 50 points or more	Changes in quality of life, depression, anxiety, gut microbiota composition and stool consistency.	Abdominal pain, diarrhea, bloating	No
Singh et al. [24]	Single-donor capsules; six separate donors (frozen)	19 capsules containing glycerol with a brown coloring agent	Oral capsules	14.25 g (single dose of 19 capsules with each pill consisting of 0.75 g)	One day	Ten weeks	IBS-SSS, IBS-QoL, IBS-GIS	Decrease in IBS-SSS ≥ 50 points	NA	NA	No (improvement in IBS symptoms but not statistically significant)

g: gram; IBS-SSS: IBS Severity Scoring System; IBS-QOL questionnaire: IBS Quality of Life questionnaire; HADS: Hospital Anxiety and Depression Scale; BSFS: Bristol Stool Form Scale; NA: not available.

**Table 4 ijms-24-14562-t004:** Intestinal microbiota modifications.

First Author	Intestinal Microbiota Modifications
Johnsen et al. [34]	*Alistipes* spp. ↑, *Bacteroides* spp. ↑, *Prevotella* spp. ↑, *Firmicutes* spp. ↑, *Akkermansia muciniphila* ↑ *Eubacterium hallii* group ↓, *Dorea* spp.↓
Halkjær et al. [37]	Clostridiales ↑, Bacteroidales ↑, biodiversity ↑
Aroniadis et al. [36]	Bacteroidetes to Firmicutes ratio ↑, *Prevotella* spp.↑
Holster et al. [39]	Butyrate-producing bacteria did not change.
Holvoet et al. [35]	NA
Lahtinen et al. [38]	Microbial richness ↑
Singh et al. [24]	None

NA: Not available.

**Table 5 ijms-24-14562-t005:** Subgroup meta-analysis of included studies.

Subgroups		Number of Studies	RR (CI 95%)	*p*-Value	I^2^	Publication Bias (*p*-Value)
FMT preparation						
	Frozen vs. non-FMT placebo	3	0.30 (0.13–0.68)	0.004	29.78	1.00
	One donor vs. autologous FMT	2	1.67 (0.59–4.67)	0.32	33.46	-
	Multiple donors vs. autologous FMT	2	2.54 (1.20–5.37)	0.01	0.00	-
	Multiple donors vs. non-FMT placebo	2	0.16 (0.05–0.48)	0.00	0.00	-
FMT frequency						
	Single dose vs. autologous FMT	4	2.20 (1.20–4.03)	0.010	0.00	0.30
	Multiple doses vs. non-FMT placebo	2	0.32 (0.07–1.32)	0.11	61.46	-
FMT route						
	Lower GI vs. autologous FMT *	4	2.20 (1.20–4.03)	0.010	0.00	0.30
	Upper GI vs. non-FMT placebo **	3	0.30 (0.13–0.68)	0.004	29.78	1.00

* Lower GI comprises colonoscopy and nasojejunal probe; ** Upper GI consists of oral administration.

## Data Availability

All data supporting the conclusions of this article are included within the article and the Appendix A.

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
