# Peer review of "Fecal Microbiota Transplantation in Irritable Bowel Syndrome: A Systematic Review and Meta-Analysis of Randomized Controlled Trials"

_ijms, 2023, doi:10.3390/ijms241914562_

Round 1

Reviewer 1 Report

Thanks to Jamshidi et al. submission of a manuscript entitled “Fecal microbiota transplantation in irritable bowel syndrome: A systematic review and Meta-analysis of Randomized Controlled Trials”, this review synthesizes current research findings and ascertains the effectiveness of FMT in relieving symptoms in IBS patients. Authors find that lower GI administration of a single dose of multiple donor FMT significantly alleviates patient complaints compared with the autologous FMT used as the placebo. The research method used by authors was reliable, and the research results were of great significance for effectively alleviating the symptoms of IBS patients.

Author Response

Dear reviewer,

Thank you very much for your positive feedback on our manuscript. It's our pleasure.

Wish you all the best,

Parnian Jamshidi

Author Response

Dear reviewer,

First and foremost, I'd like to express my profound gratitude for your comprehensive review of our manuscript. We have carefully edited the manuscript as you requested and provided a point-by-point response attached.

Reviewer 3 Report

Jamshidi et al conducted a systematic review and meta-analysis to evaluate the impact of fecal microbiota transplantation (FMT)  on irritable bowel syndrome (IBS) patients. The review involved a thorough search of PubMed/Medline and Embase databases, identifying seven studies that met the inclusion criteria. The study distinguishes itself from previous research by including the most recent relevant trials, considering both invasive and non-invasive administration routes for FMT, and accounting for differences in IBS diagnosis criteria. Their meta-analysis demonstrated no efficacy of FMT in the IBS treatment. Additionally, the text delves into the role of the gut microbiome in IBS and discusses dysbiosis (imbalance in the gut bacterial profile) as a crucial factor. The potential benefits and controversies surrounding FMT and its mechanisms are explored, including its impact on the gut-brain axis and the effects of different FMT administration routes and dosages. This review paper looks good to me. I only have some minor comments.

Minor comments:

1.     Lines 29: “From 1015 identified studies” -> “Among 1015 identified studies”

2.     Line 133: “In the primary search in mentioned databases” -> “In the primary search in the mentioned databases”

3.     Line 257: “IBS patients have low diversity and richness” -> “IBS patients have lower diversity and richness”

4.     Line 325: “Hence, the trend towards” -> “Hence, the trend toward”

The grammar looks good to me. I only have some grammatical suggestions included above.

Author Response

Dear reviewer,

Thank you very much for your thoughtful comments. We have carefully edited the manuscript as you requested.